# Accuracy, Validity, and Reliability of Markerless Camera-Based 3D Motion Capture Systems versus Marker-Based 3D Motion Capture Systems in Gait Analysis: A Systematic Review and Meta-Analysis

**DOI:** 10.3390/s24113686

**Published:** 2024-06-06

**Authors:** Sofia Scataglini, Eveline Abts, Cas Van Bocxlaer, Maxime Van den Bussche, Sara Meletani, Steven Truijen

**Affiliations:** 4D4ALL Laboratory, Department of Rehabilitation Sciences and Physiotherapy, Center for Health and Technology (CHaT), Faculty of Medicine and Health Sciences, University of Antwerp, 2000 Antwerpen, Belgium; eveline.abts@student.uantwerpen.be (E.A.); cas.vanbocxlaer@student.uantwerpen.be (C.V.B.); maxime.vandenbussche@student.uantwerpen.be (M.V.d.B.); sara.meletani@student.uantwerpen.be (S.M.); steven.truijen@uantwerpen.be (S.T.)

**Keywords:** 3D markerless camera-based motion capture, 3D marker-based motion capture, gait analysis, accuracy, reliability, validity

## Abstract

(1) Background: Marker-based 3D motion capture systems (MBS) are considered the gold standard in gait analysis. However, they have limitations for which markerless camera-based 3D motion capture systems (MCBS) could provide a solution. The aim of this systematic review and meta-analysis is to compare the accuracy, validity, and reliability of MCBS and MBS. (2) Methods: A total of 2047 papers were systematically searched according to PRISMA guidelines on 7 February 2024, in two different databases: Pubmed (1339) and WoS (708). The COSMIN-tool and EBRO guidelines were used to assess risk of bias and level of evidence. (3) Results: After full text screening, 22 papers were included. Spatiotemporal parameters showed overall good to excellent accuracy, validity, and reliability. For kinematic variables, hip and knee showed moderate to excellent agreement between the systems, while for the ankle joint, poor concurrent validity and reliability were measured. The accuracy and concurrent validity of walking speed were considered excellent in all cases, with only a small bias. The meta-analysis of the inter-rater reliability and concurrent validity of walking speed, step time, and step length resulted in a good-to-excellent intraclass correlation coefficient (ICC) (0.81; 0.98). (4) Discussion and conclusions: MCBS are comparable in terms of accuracy, concurrent validity, and reliability to MBS in spatiotemporal parameters. Additionally, kinematic parameters for hip and knee in the sagittal plane are considered most valid and reliable but lack valid and accurate measurement outcomes in transverse and frontal planes. Customization and standardization of methodological procedures are necessary for future research to adequately compare protocols in clinical settings, with more attention to patient populations.

## 1. Introduction

Gait analysis, defined as the systematic study of human gait, has developed rapidly in recent decades [1]. In various research areas, gait analysis is seen as a fundamental resource in clinical settings [2,3,4,5,6]. Three-dimensional motion capture systems are an essential tool in medical gait analysis and are being further developed to be used as efficiently as possible in the medical field [7,8,9]. Whole-body 3D gait scanning provides useful information on kinematic, kinetic, and spatial-temporal parameters [10]. The information extracted from this analysis can be used for diagnosis, analyzing gait errors, and dysfunction detection [11]. In addition, research using qualitative motion analysis equipment has also provided a great deal of insight into gait patterns [7].

Some considerations can be made in 3D motion capture systems to create the kinematic model (digital human model) [12,13,14,15]. On one hand, MBSs (marker-based motion capture systems) consist of optoelectronic cameras that record full-body motion using markers attached to body segments and are considered the gold standard in gait analysis [11,16]. However, some limitations include the time-consumption and cost, skin motion artifacts, close contact with the patients, and uncomfortable situation limiting the performance of the user [11,17]. This can be perceived as an intrusive procedure for the patient and may also affect the naturalness of movement and make it difficult to use in clinical gait assessment laboratories [11,17]. In the quest for more efficient and less intrusive motion capture techniques, markerless camera-based systems have emerged as promising alternatives to traditional 3D marker-based methods. These markerless systems, including 3D (static) and 4D (dynamic, as 3D + time) scanners, allow for the capture of the full body shape of the subject [18,19], offering several advantages over their marker-based counterparts, addressing key limitations, and opening new avenues for motion capture and analysis [11,18,20].

One significant advantage of markerless systems is their potential to mitigate the time-consuming and costly nature of marker-based approaches [8,11]. By eliminating the need to attach markers to body segments, markerless systems streamline the setup process, reducing both setup time and associated costs. This streamlined workflow can be particularly advantageous in clinical settings where efficiency and cost-effectiveness are paramount. Moreover, markerless systems reduce error related to skin motion artifacts inherent in marker-based approaches. Since markerless systems capture motion directly from the subject’s surface geometry, they are less susceptible to inaccuracies caused by marker movement or slippage [11,18,20]. This enhanced accuracy can lead to more reliable motion capture data, crucial for applications such as gait analysis and biomechanical research.

Additionally, markerless systems offer a non-intrusive alternative to marker-based methods, eliminating the need for physical contact with the patient’s body. This non-intrusiveness not only enhances patient comfort but also preserves the naturalness of movement during data capture. By reducing discomfort and promoting natural movement, markerless systems can yield more ecologically valid motion data, better reflecting real-world scenarios [18].

However, despite these advantages, markerless systems still face challenges to the gold standard set by marker-based motion capture. Research efforts are ongoing to enhance the performance of markerless systems, addressing issues such as occlusions, noise, and computational complexity. Advancements in computer vision algorithms, machine learning techniques, and sensor technologies are driving improvements in markerless motion capture, bringing them closer to parity with marker-based methods [18].

A fundamental aspect of these systems is that they must capture accurate, valid, and reliable data, especially where we are assessing gait analysis for early disorder (e.g., neurological diseases) identification and monitoring in medicine [21].

However, at this moment, in literature, there is no clear consensus yet regarding the accuracy, validity, and reliability of the markerless systems in gait analysis [11,22]. Maynard et al. [16] even stated a lack of consistency in the reported results. Since this gap is present, this systematic review aims to compare the accuracy, validity, and reliability of 3D markerless camera-based motion capture systems against 3D marker-based motion capture systems in full-body gait analysis. Further, a quantitative synthesis of gait parameters (e.g., spatial, temporal, and kinematic) will be conducted to measure whether the 3D markerless camera-based gait analysis systems are comparable to the 3D marker-based motion capture systems in terms of accuracy, validity, and reliability.

In this systematic review and meta-analysis, a definition of psychometric properties (accuracy, validity, and reliability) is necessary for a proper understanding of what is being studied, as some psychometric properties may have different meanings depending on the context or may be used as synonyms for each other. Accuracy refers to how close the values of a given system are to the standard against which it has been measured [23]. Accuracy in this context refers to absolute agreement, as we are looking for differences that have been measured. Validity in this systematic review refers to concurrent validity [23], where two methods are compared simultaneously when measuring relationships between variables. Correlations between the two methods are calculated, with high correlations between the two systems confirming concurrent validity [23]. In this context, validity refers to relative agreement. Three different definitions of reliability are used in this systematic review. Inter-trial reliability, inter-rater reliability, and within-session/intra-session reliability are considered. Inter-trial reliability refers to the test-retest reliability of how stable measurements are when conditions remain unchanged over time [23]. Inter-rater reliability refers to the consistency of measurements made by different raters/systems. Intra-session reliability refers to the consistency of measurements made by the same rater/system in a series of measurements made under the same conditions [23].

## 2. Methods

This systematic review and meta-analysis, with the aim to compare the accuracy, concurrent validity, and reliability of markerless camera-based 3D motion capture systems against marker-based 3D motion capture systems in full-body gait analysis, adhered to the guidelines of Preferred Reporting Items for Systematic reviews and Meta-Analyses (PRISMA) [24].

### 2.1. Data Sources and Search Strategy

A systematic literature research was carried out to retrieve articles concerning 3D markerless camera-based versus marker-based 3D motion capture systems in gait analysis on the electronic databases, PubMed and Web of Science (WoS), on 7 February 2024.

The search strategy used in PubMed (Table 1) based on the PICO strategy, consisted of keywords and Medical Subject Headings (MeSH), to make the search as comprehensive as possible. Keywords that did not provide more search results were carefully ruled out. In addition, the search strategy was adapted to Web of Science (Table 2). However, no filters were applied in this database because the filter used in PubMed is not available in Web of Science. The full search string used for PubMed and Web of Science is shown in Table 1 and Table 2, respectively.

### 2.2. Study Selection

The outcomes of the combined searches from PubMed and Web of Science were integrated in Endnote, followed by a manual removal of duplicates.

Inclusion and exclusion criteria were carefully used to determine if an article was relevant or not for this research to generate an appropriate answer to the previously stated research question. The predefined inclusion and exclusion criteria are consistent with the research question and can be found in Table 3.

The search was limited to literature reporting studies of human adults with abstracts written in English or Dutch (Table 3). Only articles that describe gait analysis performed by a marker-less camera-based 3D and 4D motion capture system of lower body (pelvis, hip, knee, and ankle) and marker-based 3D motion capture systems of lower body (pelvis, hip, knee, and ankle) were considered. In addition, gait performed overground or on a treadmill was allowed indoors, considering only walking speed <1.8 m/s as the threshold to distinguish between walking and running [25].

The exclusion of 2D markerless camera or pose estimation methods, particularly in contexts involving multiple cameras for 3D reconstruction, is often due to several factors. These 2D systems lack depth information, relying on visual cues susceptible to occlusions and lighting variations, leading to reduced accuracy [12]. Multi-camera markerless setups for 2D pose estimation require complex algorithms for calibration and triangulation, increasing computational cost. In contrast, 3D motion capture systems offer a more direct approach with higher accuracy and efficiency in capturing three-dimensional motion [12]. Precise measurement of 3D motion is essential in applications like biomechanics and gait analysis, where 2D methods may not suffice. Advancements in 3D technology, including marker-based and markerless systems, further emphasize the preference for comprehensive motion capture solutions over traditional 2D approaches [12].

All articles were evaluated for eligibility in a two-stage screening procedure. The first screening was based on title, abstract, inclusion, and exclusion criteria. Every article was screened at least twice independently by three authors (EA, CVB, MVDB) following the PICO method. This process was distributed to ensure that each article was screened double-blindly by two researchers. By using this distribution, a third person could be assigned to review an article. When a disagreement occurred, a consensus was reached by that third author who made the conclusive decision. The screening took place in Rayyan, an online software program, to guarantee the blinding procedure. Articles that did not meet the eligibility criteria were excluded. The second screening was double-blinded by three researchers (EA, CVB, MVDB). In this stage of the screening process, the full text of the remaining articles was rated, applying the inclusion and exclusion criteria. The blinded procedure was applied to ensure that there was no influence on the researcher’s judgement. Conflicts were resolved similarly to stage one conflicts, with the use of a third author. The screening of the manually searched articles went through the same screening procedure as the articles from the search strategy.

### 2.3. Data Collection Process

The outcome of the combined database search string in PubMed and Web of Science resulted in 2.047 potential records, as shown in Figure 1. After the first screening, 143 studies were found relevant for the second selection. During the full-text screening of those studies, 121 articles were excluded based on either population, intervention, comparison, outcome, or language. Twenty-two articles have been considered eligible from the literature screening process to provide an answer to the research question concerning reliability, validity, and accuracy in 3D camera-based markerless and 3D marker-based motion capture systems. The procedure for the selection of the articles and the reasons for their exclusion can be found in Figure 1.

### 2.4. Data Extraction

#### 2.4.1. Qualitative Systematic Review

A total of 22 relevant studies were retained for inclusion in this systematic review after the screening process. The extraction of the outcome data was performed and discussed by three authors (EA, CVB, MVDB) for study design/level of evidence, participant characteristics, test condition, gait speed (m/s), measuring method, reference method, sampling rate (Hz), gait parameters, type of reliability, concurrent validity metrics, and accuracy metrics. A brief summary of all the above-mentioned characteristics of the articles is given in Table 4. The findings of the included articles on concurrent validity, accuracy, and reliability type are summarized in Table 5. They are given for each gait parameter (e.g., hip, knee, and ankle kinematics; spatial, temporal, and walking speed).

According to the COSMIN guidelines, intra-class correlation coefficient (ICC) selection is a valid reliability measure [26,27,28]. For this reason, only consistency intra-class correlations coefficients (C-ICC) were retained for reporting in this review in terms of reliability. The interpretation of these results is reported in relation to the study of Koo and Li [29]. ICC values for reliability and validity are interpreted in the following manner: poor (<0.50), moderate (0.50–0.75), good (0.75–0.90), and excellent (>0.90) [29].

The data for concurrent validity in this context relative agreement are expressed as Pearson’s correlation coefficients (r), (Lin’s), coefficient of multiple correlations (CMC), and agreement intra-class correlation coefficient (A-ICC).

Pearson’s correlation was interpreted in the same manner: poor (<0.40), modest (0.40–0.74), or excellent (>0.75). The concordance coefficient measures the overall agreement, and both precision and deviation are considered, while Pearson’s correlation assesses precision (relative agreement) [30].

The data for accuracy are expressed in this article as bias, error (°), RMSE (root mean square error), RMS (root mean square), and RMSD (root mean square deviation). Smaller errors indicate better accuracy. In this context, absolute agreement was verified by looking at the differences between the systems.

For this review, spatial parameters include step length, step width, and stride length. Temporal parameters include step time, stride time, stance time, and swing time. Gait speed was considered a separate parameter because this parameter implements both spatial and temporal factors.

**Table 4 sensors-24-03686-t004:** Table of evidence for reliability, concurrent validity, and accuracy.

Study, (Year)	Design/Level of Evidence	Participants Characteristics n, Age (Years), Gender Healthy/Patients	Test Conditions, Speed (m/s)	Measuring Method, Sampling Rate	Reference Method, Sampling Rate (Hz)	Gait Parameters	Type of Reliability; Reliability Metric	Concurrent Validity Metric	Accuracy Metric
Clark et al. (2013) [31]	Cross-sectional/B	n = 21, 26.9 (±4.5), 11 F; 10 M Healthy	Overground, self-selected speed	Markerless Microsoft Kinect, 30 Hz	Marker-based Vicon Nexus V1.5.2 with 12 Vicon MX cameras (Vicon, UK), 120 Hz	Step length (m), stride length (m), stride time (s), step time (s), walking speed (m/s)	Inter-rater reliability rc 95% CI (CCC)	r	
Arango Paredes et al. (2015) [32]	Case study/C	n = 1, 34, 1 F Healthy	Overground, self-selected speed	Markerless (E motion capture system based on Kinect TM), 60 Hz	Marker-based (Multiple-camera 3D motion capture system), 120 Hz	Cadence (steps/min), stride length (m), step length (m), step width (m), walking speed (m/s),	Inter-rater reliability, ICC (3, 2)		
Eltoukhy et al. (2017) [33]	Cross-sectional/B	n = 11, 71.1 (± 7.5), Healthy n = 8, 71.0 (± 5.6), PD	Overground, self-selected speed (1.06 m/s), barefoot	Markerless Kinect V2, 30 Hz	Marker-based 8-camera BTS optoelectronic motion capture system, 100 Hz	ROM (°) sagittal plane (ankle, knee, hip), stride length (m), stance duration (s), swing duration (s), cadence (step/min), swing velocity (m/s), walking speed (m/s)	Inter-rater reliability, ICC (2, k) ± CI 95% (combined sample)	A-ICC	
Eltoukhy et al. (2017) [34]	Cross-sectional/B	n = 10, 26.7 (±5.4), 5 F, 5 M Healthy	Treadmill, 1.3 m/s and 1.6 m/s	Markerless A single Kinect v2 sensor (Microsoft Corp. Redmond, WA), 30 Hz	Marker-based An eight infrared camera motion analysis system (SMART-DX 7000, BTS Bioengineering, Milano, Italy), 100 Hz	ROM (°) sagittal plane (ankle, knee, hip), stride time (s), step time (s), step width (m), step length (m)	Inter-rater reliability, ICC (2, k)	A-ICC r	
Ripic et al. (2022) [35]	Cross-sectional/B	n = 22, 22.72, F12; M10 Healthy	Overground Usual Pace	Markerless eight video cameras (KinaTrax Inc., Boca Raton, FL, USA 100 Hz	Marker-based, Vicon Plug-in-Gait (Vicon Motion Systems Inc., Oxford, UK), 100 Hz	Walking speed (m/s), stride length (m), stride width (m), step length (m), cycle time (s), stance time (s), swing time (s), step time (s), double limb support (s)	Inter-rater reliability Consistency ICC (2, k)	A-ICC	
Mentiplay et al. (2015) [36]	Control trial/B	n = 30, 22.87 (±8 5.08), 15 F; 15 M Healthy	Overground, comfortable speed (1.26 m/s) and fast-paced trials (1.63 m/s)	Markerless Microsoft Kinect V2, 30 Hz	MCBS Vicon (9 camera Vicon system), 100 Hz	ROM (°) sagittal plane (hip, knee, ankle), walking speed (m/s), step length (m), step time (s), step width (m), foot swing velocity (m/s)	Within system, Inter-trial reliability (2 sessions 7 days apart), ICC (2, 1)	r	
Dolatabadi et al. (2016) [37]	Cross-sectional/B	n = 20, 28.8, F10; M10 Healthy	Overground Usual pace	Kinect V2 30 Hz	Gait rite 120 Hz	Stance time (s) Step time (s) Step length (m) Walking speed (m/s)	Inter-trial reliability, ICC (3, 1)		
Fosty et al. (2016) [38]	Cross-sectional/B	n = 36, 32.1 (±7.6), 19 F; 17 M Healthy	Treadmill, 0.42 m/s, 0.69 m/s, 0.97 m/s, 1.25 m/s, 1.53 m/s	Markerless Asus1 Xtion PRO LIVE RGB-D camera + Kinect sensor (PCBS)	Marker-based control system (MBCS)	Walking speed (m/s)	Intra-session reliability, ICC (2, 1)		Bias
Otte et al. (2016) [39]	Cross-sectional/B	n = 19, 29.5 (±4.4), 12 F; 7 M Healthy	Overground, Comfortable speed walk (Mean: 1.29 m/s)	Markerless Motognosis Labs v1.0 (Motognosis UG, Berlin, Germany) with a Kinect for Windows V2 Sensor (25 (−4) markers), 30 Hz	Marker-based 1 6-camera Vicon system (MX13+, Nexus 2.1; Vicon Motion Systems Ltd., Oxford, UK) (36 IR reflecting markers), 100 Hz	Mean walking speed (m/s)	Intra-session reliability, ICC (1, 1)		
Pfister A et al. (2014) [40]	Cross-sectional/B	n = 20, 27.4 (±10), 11 F; 9 M Healthy	Treadmill, 1.34 m/s	Markerless Xbox Kinect (Microsoft, Redmond, WA, USA), 30–37 Hz	Marker-based Vicon MX motion capture system (Vicon, Oxford, UK), 120 Hz	ROM sagittal plane (°) (hip, knee) Stride time (s)		r	Error
Timmi et al. (2018) [41]	Cross- sectional/B	n = 20, 31 (±6), 9 F; 11 M Healthy	Treadmill Slow walking speed (0.83 m/s) Fast walking speed (1.31 m/s)	Markerless Kinect V2- based system 30 Hz	Marker-based 12-camera Vicon motion capture 120 Hz	ROM (°) frontal, sagittal, transverse plane (knee, ankle, hip)			Bias (°) (Upper LOA (°), Lower LOA (°))
Xu et al. (2015) [42]	Cross-sectional/B	n = 20, 28.5 (± 8.2), 10 F; 10 M Healthy	Treadmill 3 walking speeds: 0.85 m/s, 1.07 m/s, 1.30 m/s	Markerless Kinect sensor (Kinect for windows SDK 1.5) 60 Hz	Marker-based A motion tracking system (Optotrack Certus System, Northern Digital, Canada), 60 Hz	ROM (°) frontal plane (hip, knee), step time (s), stride time (s), stance time (s), swing time (s), double support time (s), step width (m)		r	RMSE (°)
Kanko et al. (2021) [43]	Cross-sectional/B	n = 30, 23.0 (±3.50), 15 F; 15 M Healthy	Treadmill Comfortable self-selected speed	Markerless Theia3D (Theia Markerless Inc., Kingston, ON, Canada), 85 Hz	Marker-based camera system (seven Qualisys 3+ (Qualisys AB, Gothenburg, Sweden), 85 Hz	The average 3D Euclidean distance between corresponding limb joints Lower limb segment angles (m)			RMSD (cm) RMSD (°)
Kanko et al. (2021) [44]	Cross-sectional/B	n = 30, 23.0 (± 3.50), 15 F; 15 M Healthy	Treadmill Start at 1.2 m/s, self-selected speed	Markerless Theia3D (Theia Markerless Inc., Kingston, ON, Canada), 85 Hz	Marker-based camera system (seven Qualisys 3+ (Qualisys AB, Gothenburg, Sweden) 85 Hz	Gait speed (m/s), step length (m), stride length (m), step width (m), step time (s), cycle time (s), swing time (s), stance time (s), double limb support time (s)		ICC-A,1 ICC (LB-UB) r	
Albert et al. (2020) [45]	Pilot study/B	n = 5, 28.4 (±4.20) Healthy	Treadmill Walking speed of 0.83 m/s, 1.07 m/s, 1.30 m/s	Markerless Azure Kinect and Kinect V2 30 Hz	Marker-based Vicon multi-camera motion capturing system and the 39 marker Plug-in Gait model, 100 Hz	ROM (°) frontal, sagittal, and transverse plane (hip, knee, ankle), step length (m), step width (m), step time (s), stride time (s)		r	RMSE AE
Vilas-Boas et al. (2019) [46]	Cross-sectional/B	n = 20, 30.5 (±8.07), 10 F; 10 M Healthy	Overground, self-selected comfortable speed	Markerless Kinect v1 and Kinect v2 (Microsoft Corporation, Redmond, WA, USA), 30 Hz	Marker-Based Qualisys system (Qualisys AB, Sweden), 200 Hz	ROM (°) frontal plane (hip, knee, ankle), walking speed (m/s)		r	
Tanaka et al. (2018) [47]	Cross-sectional/B	n = 51, 20.9 (± 0.2), 16 F; 35 M Healthy	Overground Comfortable speed	Markerless A Kinect v2 sensor (Microsoft Corporation, Redmond, WA, USA) (frequency not mentioned)	Marker-based Vicon Motion Systems, Oxford, UK 120 Hz	Gait cycle sagittal & frontal angles (°) (Hip, knee)		r	
Dubois & Bresciani (2018) [48]	Cross-sectional/B	Young: n = 8, 25, 5 F; 3 M Older: n = 9, 69, 5 F; 4 M Senior: n = 8, 81, 5 F; 3 M Healthy	Overground Comfortable speed	Markerless A single Microsoft Kinect V2 camera	Marker-based OptiTrack cameras (Prime 17 W model)	Step length (m), step duration(s), cadence (steps/min), walking speed (m/s)		A-ICC	
Muller et al. (2017) [49]	Cross-sectional/B	n = 10, 18–35 Healthy	Overground Comfortable speed	Markerless Microsoft Kinect V2 30 Hz	Marker-based Vicon MX motion capture system 120 Hz	Walking speed (m/s), step time (s), stride length (m), step length (m), step width (m)		ICC (A,1) r	
Ruescas Nicolau et al. (2022) [50]	Cross-sectional/B	n = 12, 39.1 (± 9.8), 5 F; 7 F Healthy	Overground Comfortable speed	Markerless 3 D temporal scanner with 16 camera modules (Move4D/IBV), 30 fps	Marker- based stereophotogrammetry system (Kinescan/IBV) (16 cameras) 30 fps	Joint angle (°) frontal, sagittal, and transverse plane errors (hip, knee)			RMS
Ma et al. (2020) [51]	Cross-sectional/B	n = 5, 29.8 (±5.8), 3 F; 2 M Healthy	Overground comfortable speed	Markerless Dual Azure Kinect- Based motion capture system, 30 Hz	Marker-based Eight camera based Vicon Motion capture system (Oxford Metrics Group, Oxford, UK), 100 Hz	ROM (°) sagittal and frontal plane (hip, knee, ankle)			CMC RMSE (°)
Ripic et al. (2023) [52]	Cross-sectional/B	Young: n = 17, 21 (± 2), 5 F; 12 M Older: n = 7, 74 (±5), 3 F; 4 M PD: n = 11, 70 (± 8), 5 F; 6 M	Overground, usual speed	8-camera markerless (KinaTrax Inc., Boca Raton, FL, USA), 100 Hz	Vicon Motion Systems Inc., Oxford, UK, 100 Hz	ROM (°) (hip, knee, and ankle) in sagittal, frontal, and transverse plane		ICC (3, 1)r	RMSE ± SD

n: number of participants, m/s: meter per second, Hz: hertz, F: female, M: male, 3D: three dimensional, CCC: consistency correlation coefficient, r: Pearson’s correlation coefficient, ICC: intraclass correlation coefficient, A-ICC: agreement intraclass correlation coefficient, ROM: range of motion, MBCS: marker-based control system, PCBS: point cloud-based system, LOA: limit of agreement, RMSE: root mean square error, RMSD: root mean square deviation, cm: centimeter, °: degree, LB: lower body, UB: upper body, AE: absolute error, RMS: root mean square, CMC: coefficient of multiple correlations, SD: standard deviation.

**Table 5 sensors-24-03686-t005:** Evidence table: Results of all included articles for the systematic review.

	Study, Year	Inter-Rater Reliability	Inter-Trial Reliability	Intra-Session Reliability	Concurrent Validity	Accuracy
Kinematics Hip	Eltoukhy et al. (2017) [33] Healthy	C-ICC: 0.92; 0.98			sagittal plane A-ICC: 0.86; 0.94	
	Eltoukhy et al. (2017) [33] Parkinson	C-ICC: 0.93; 0.98			sagittal plane A-ICC: 0.94; 0.97	
	Eltoukhy et al. (2017) [33] Combined	C-ICC: 0.94; 0.96			sagittal plane A-ICC: 0.92; 0.94	
▸	Eltoukhy et al. (2017) [34] 1.3 m/s	C-ICC: 0.85 (0.38; 0.96)			A-ICC: 0.77 (0.05; 0.95) r: 0.73 (0.17; 1.25)	
▸	Eltoukhy et al. (2017) [34] 1.6 m/s	C-ICC: 0.86 (0.45; 0.97)			A-ICC: 0.80 (0.10; 0.95) r: 0.77 (0.26; 1.42)	
	Mentiplay et al. (2015) [36] Kinect comfortable speed		C-ICC: −0.10 (−0.57; 0.42)			
	Mentiplay et al. (2015) [36] Kinect fast-paced		C-ICC: 0.23 (−0.40; 0.71)			
	Mentiplay et al. (2015) [36] Vicon comfortable speed		C-ICC: 0.55 (0.16; 0.80)			
	Mentiplay et al. (2015) [36] Vicon fast-paced		C-ICC: 0.34 (−0.05; 0.64)			
	Pfister A et al. (2014) [40] ROM sagittal plane				r: −0.04; 0.27 *	Hip angular displacement was poor (r < 0.30) with errors greater than 5° in every case
▸	Xu et al. (2015) [42] 0.85 m/s					RMSE: 11.8 (8.6)
▸	Xu et al. (2015) [42] 1.07 m/s					RMSE: 11.7 (8.6)
▸	Xu et al. (2015) [42] 1.30 m/s					RMSE: 11.9 (8.9)
▸	Albert et al. (2020) [45] Ki V2				AP: r 0.98 ML: r 0.95; 0.96 V: r 0.78; 0.80	
▸	Albert et al. (2020) [45] Azure Ki				AP: r 0.98; 0.99 ML: r 0.89; 0.91 V: r 0.60; 0.68	
	Vilas-Boas et al. (2019) [46] walking towards sensor				Ki V1: r 0.62 (0.39; 0.85) Ki V2: r 0.13 (−0.20; 0.46)	
	Vilas-Boas et al. (2019) [46] walking away from sensor				Ki V1: r 0.13 (−0.29; 0.55) Ki V2: r 0.54 (0.28; 0.80)	
	Tanaka et al. (2018) [47] sagittal plane				r: 0.43; 0.78 *	
	Tanaka et al. (2018) [47] frontal plane				r: 0.09; 0.71	
	Ruescas Nicolau et al. (2022) [50] FE					RMS: 1.26 ± 0.3 (3.0%)
	Ruescas Nicolau et al. (2022) [50] LF					RMS: 1.65 ± 0.44 (13.8%)
	Ruescas Nicolau et al. (2022) [50] AR					RMS: 5.76 ± 1.95 (43.3%)
	Ma et al. (2020) [51] sagittal and frontal plane				CMC: 0.48; 0.60	RMSE: 7.2°; 15.1°
	Ma et al. (2020) [51] transverse plane				Internal & external rotation hip CMC < 0.001	RMSE: 32.2° ± 22.2
	Ripic et al. (2023) [52] sagittal plane				ICC (Cl 95%): 0.98 (0.98; 0.98) r: 0.99	RMSE: 8.21 ± 4.06
	Ripic et al. (2023) [52] frontal plane				ICC (Cl 95%): 0.49 (0.45; 0.53) r: 0.56	RMSE: 3.16 ± 1.30
	Ripic et al. (2023) [52] transverse plane				ICC (Cl 95%): 0.07 (0.03; 0.12) r: 0.09	RMSE: 11.90 ± 3.83
	Eltoukhy et al. (2017) [33] Healthy	C-ICC: 0.69; 0.96			A-ICC: 0.70; 0.92	
Kinematics Knee	Eltoukhy et al. (2017) [33] Parkinson	C-ICC: 0.92; 0.97			A-ICC: 0.93; 0.98	
	Eltoukhy et al. (2017) [33] Combined	C-ICC: 0.93; 0.96			A-ICC: 0.90; 0.96	
▸	Eltoukhy et al. (2017) [34] 1.3 m/s	C-ICC: 0.66 (−0.39; 0.91)			A-ICC: 0.68 (−0.45; 0.92) r: 0.57 (−0.17; 2.21)	
▸	Eltoukhy et al. (2017) [34] 1.6 m/s	C-ICC: 0.82 (0.27; 0.96)			A-ICC: 0.80 (0.26; 0.95) r: 0.75 (0.32; 1.95)	
	Mentiplay et al. (2015) [36] Kinect comfortable speed		C-ICC: 0.69; 0.85			
	Mentiplay et al. (2015) [36] Kinect fast-paced		C-ICC: 0.38; 0.75			
	Mentiplay et al. (2015) [36] Vicon comfortable speed		C-ICC: 0.58; 0.91			
	Mentiplay et al. (2015) [36] Vicon fast-paced		C-ICC: 0.55; 0.86			
▸	Pfister A et al. (2014) [40] ROM sagittal plane				r: 0.43; 0.77 *	
▸	Timmi et al. (2018) [41] slow walk					knee flexion bias (upper LOA; lower LOA): −0.1 (−1.1; 1.0) knee adduction bias (upper LOA; lower LOA): −0.2 (−1.6; 1.3)
▸	Timmi et al. (2018) [41] fast walk					knee flexion bias (upper LOA; lower LOA): 0.0 (−0.7; 0.8) knee adduction bias (Upper LOA; lower LOA): −0.6 (−2.8; 1.7)
▸	Xu et al. (2015) [42] 0.85 m/s					RMSE: 27.9 (10.0)
▸	Xu et al. (2015) [42] 1.07 m/s					RMSE: 28.6 (10.8)
▸	Xu et al. (2015) [42] 1.30 m/s					RMSE: 29.0 (10.3)
▸	Albert et al. (2020) [45] Ki V2				AP: r 0.98 ML: r 0.93; 0.95 V: r 0.35; 0.41	
▸	Albert et al. (2020) [45] Azure Ki				AP: r 0.97; 0.98 ML: r 0.87; 0.94 V: r 0.73; 0.74	
	Vilas-Boas et al. (2019) [46] walking towards sensor				Ki V1: r 0.93 (0.82; 1.00) Ki V2: r 0.94 (0.84; 1.00)	
	Vilas-Boas et al. (2019) [46] walking away from sensor				Ki V1: r 0.87 (0.69; 1.00) Ki V2: r 0.91 (0.71; 1.00)	
	Tanaka et al. (2018) [47] sagittal plane				r: 0.49; 0.88 *	
	Tanaka et al. (2018) [47] frontal plane				r: 0.50; 0.90 *	
	Ruescas Nicolau et al. (2022) [50] FE					RMS: 1.98 ± 0.37 (3.3%)
	Ruescas Nicolau et al. (2022) [50] LF					RMS: 3.51 ± 1.23 (37.1%)
	Ruescas Nicolau et al. (2022) [50] AR					RMS: 3.62 ± 1.34 (14.8%)
	Ma et al. (2020) [51] frontal and sagittal plane				CMC: 0.87 (± 0.06)	RMSE: 11.9° ± 3.4
	Ripic et al. (2023) [52] sagittal plane				ICC (Cl 95%): 0.99 (0.99; 0.99) r: 0.99	RMSE: 7.97 ± 2.73
	Ripic et al. (2023) [52] frontal plane				ICC (Cl 95%): 0.18 (0.10; 0.26) r: 0.20	RMSE: 6.01 ± 1.40
	Ripic et al. (2023) [52] transverse plane				ICC (Cl 95%): 0.11 (0.06; 0.17) r: 0.13	RMSE: 10.42 ± 4.34
	Eltoukhy et al. (2017) [33] Healthy	C-ICC: 0.00; 0.20			A-ICC: 0.00; 0.20	
Kinematics Ankle	Eltoukhy et al. (2017) [33] Parkinson	C-ICC: 0.00; 0.17			A-ICC: 0.00; 0.17	
	Eltoukhy et al. (2017) [33] Combined	C-ICC: 0.00; 0.13			A-ICC: 0.00; 0.14	
▸	Eltoukhy et al. (2017) [34] 1.3 m/s	C-ICC: 0.01 (−0.06; 0.20)			A-ICC: 0.05 (−2.84; 0.76) r: 0.03 (−1.78; 1.93)	
▸	Eltoukhy et al. (2017) [34] 1.6 m/s	C-ICC: −0.39 (−4.60; 0.65)			A-ICC: −0.03 (−0.09; 0.19) r: −0.22 (−2.20; 1.26)	
	Mentiplay et al. (2015) [36] Kinect comfortable speed		C-ICC: 0.42 (0.02; 0.71)			
	Mentiplay et al. (2015) [36] Kinect fast-paced		C-ICC: 0.44 (−0.04; 0.75)			
	Mentiplay et al. (2015) [36] Vicon comfortable speed		C-ICC: 0.75 (0.51; 0.88)			
	Mentiplay et al. (2015) [36] Vicon fast-paced		C-ICC: 0.68 (0.38; 0.85)			
▸	Albert et al. (2020) [45] Ki V2				AP: r 0.97 ML: r 0.95; 0.97 V: r 0.76; 0.78	
▸	Albert et al. (2020) [45] Azure Ki				AP: r 0.96; 0.97 ML: r 0.81; 0.85 V: r 0.84; 0.89	
	Vilas-Boas et al. (2019) [46] walking towards sensor				Ki V1: r −0.18 (−0.43; 0.07) Ki V2: r −0.15 (−0.39; 0.09)	
	Vilas-Boas et al. (2019) [46] walking away from sensor				Ki V1: r 0.01 (−0.23; 0.25) Ki V2: r −0.02 (−0.23; 0.19)	
	Ma et al. (2020) [51] frontal and sagittal plane				CMC: 0.55 (± 0.09)	RMSE: 11.6° ± 2.4
	Ripic et al. (2023) [52] sagittal plane				ICC (Cl 95%): 0.92 (0.91; 0.93) r: 0.95	RMSE: 4.96 ± 1.84
	Ripic et al. (2023) [52] frontal plane				ICC (Cl 95%): 0.44 (0.39; 0.48) r: 0.47	RMSE: 6.01 ± 1.40
	Ripic et al. (2023) [52] transverse plane				ICC (Cl 95%): 0.10 (0.05; 0.16) r: 0.10	RMSE: 10.15 ± 3.49
▸	Kanko et al. (2021) [43]					The average RMSD (cm) between corresponding joint centers: <2.5 cm (except for the hip: 3.6 cm) RMSD: <5.5° (except those that represent rotations about the long axis of the segment)
Kinematics lower limb angles	Eltoukhy et al. (2017) [33] Healthy	C-ICC: 0.99 (0.99; 1.00)			A-ICC: 0.99 (0.99; 1.00)	
Spatial	Eltoukhy et al. (2017) [33] Parkinson	C-ICC: 0.99 (0.99; 1.00)			A-ICC: 0.99 (0.99; 1.00)	
	Eltoukhy et al. (2017) [33] Combined	C-ICC: 0.99 (0.99, 0.99)			A-ICC: 0.99 (0.99; 0.99)	
	Clark et al. (2013) [31]	CCC: 0.97; 0.99			r: 0.99 *	
▸	Eltoukhy et al. (2017) [34] 1.3 m/s	C-ICC: 0.58; 0.94			A-ICC: 0.76; 0.84 r: 0.73; 0.93	
▸	Eltoukhy et al. (2017) [34] 1.6 m/s	C-ICC: 0.87; 0.95			A-ICC: 0.67; 0.71 r: 0.84; 0.91	
	Ripic et al. (2022) [35]	C-ICC: 0.865; 0.938			A-ICC: 0.867; 0.939	
	Mentiplay et al. (2015) [36] comfortable speed		Ki: ICC 0.71; 0.87 Vi: ICC 0.79; 0.85		r: 0.90; 0.94	
	Mentiplay et al. (2015) [36] fast-paced		Ki: ICC 0.74; 0.94 Vi ICC 0.78; 0.85		r: 0.92; 0.95	
	Dolatabadi et al. (2016) [37] usual pace		C-ICC: 0.94			
▸	Xu et al. (2015) [42] 0.85 m/s				r: 0.85	
▸	Xu et al. (2015) [42] 1.07 m/s				r: 0.82	
▸	Xu et al. (2015) [42] 1.30 m/s				r: 0.79	
▸	Kanko et al. (2021) [44]				A-ICC: 0.92; 0.98 A-ICC LB: 0.90; 0.97 A-ICC UB: 0.93; 0.98 r: 0.92–0.94	
▸	Albert et al. (2020) [45] Ki V2					0.83 m/s: RMSE: 0.07 1.07 m/s: RMSE: 0.07; 0.08 1.30 m/s: RMSE: 0.06; 0.08
▸	Albert et al. (2020) [45] Azure Ki					0.83 m/s: RMSE: 0.03; 0.05 1.07 m/s: RMSE: 0.04; 0.05 1.30 m/s: RMSE: 0.04; 0.05
	Dubois & Bresciani (2018) [48]				A-ICC: 0.97	
	Muller et al. (2017) [49] One-sided				ICC (A,1): 0.882; 0.996 r: 0.944; 0.998 *	
	Muller et al. (2017) [49] Two- sided				ICC (A,1): 0.910; 0.999 r: 0.936; 0.999 *	
	Eltoukhy et al. (2017) [33] Healthy	C-ICC: 0.75; 0.99			A-ICC: 0.76; 1.00	
Temporal	Eltoukhy et al. (2017) [33] Parkinson	C-ICC: 0.68; 0.99			A-ICC: 0.72; 1.00	
	Eltoukhy et al. (2017) [33] Combined	C-ICC: 0.93; 0.99			A-ICC: 0.84; 100	
	Clark et al. (2013) [31]	CCC: 0.14; 0.23			r: 0.69; 0.82 *	
▸	Eltoukhy et al. (2017) [34] 1.3 m/s	C-ICC: 0.96; 0.98			A-ICC: 0.87; 0.98 r: 0.96; 0.97	
▸	Eltoukhy et al. (2017) [34] 1.6 m/s	C-ICC: 0.93; 0.97			A-ICC: 0.82; 0.94 r: 0.93; 0.95	
	Ripic et al. (2022) [35]	C-ICC: 0.923; 0.962			A-ICC: 0.834; 0.951	
	Mentiplay et al. (2015) [36] comfortable speed		Ki: ICC 0.03; 0.70 Vi: ICC 0.21; 0.71		r: 0.91; 0.92	
	Mentiplay et al. (2015) [36] fast-paced		Ki: ICC 0.43; 0.87 Vi: ICC 0.47; 0.90		r: 0.88; 0.94	
	Dolatabadi et al. (2016) [37] usual pace		C-ICC: 0.90; 0.92			
	Pfister A et al. (2014) [40]				r: 0.87; 0.92 *	
▸	Xu et al. (2015) [42] 0.85 m/s				r: 0.24; 0.85	
▸	Xu et al. (2015) [42] 1.07 m/s				r: 0.24; 0.92	
▸	Xu et al. (2015) [42] 1.30 m/s				r: 0.20; 0.95	
▸	Kanko et al. (2021) [44]				A-ICC: 0.82; 0.93 (outliers swing time ICC: 0.39 and double limb support time ICC: 0.53) A-ICC LB: 0.70; 0.92 (outliers swing time ICC: 0.22 and double support time ICC: 0.11) A-CC UB: 0.79; 0.94 (outlier swing time ICC: 0.53) r: 0.73; 0.93 (outlier swing time 0.47)	
▸	Albert et al. (2020) [45] Ki V2					For all velocities: RMSE 0.03
▸	Albert et al. (2020) [45] Azure Ki					For all velocities: RMSE: 0.02; 0.03
	Dubois & Bresciani (2018) [48]				ICC: 0.94	
	Muller et al. (2017) [49]				ICC (A,1): 1.000 r: 1.000 *	
Spatio-temporal	Arango Paredes et al. (2015) [32]	Average ICC: 0.96 (CI 95% 0.94; 0.97)				
Walking speed	Eltoukhy et al. (2017) [33] Healthy	C-ICC: 0.99 (0.99; 1.00)			A-ICC: 0.99 (0.99; 1.00)	
	Eltoukhy et al. (2017) [33] Parkinson	C-ICC: 1.00 (0.99; 1.00)			A-ICC: 1.00 (0.99; 1.00)	
	Eltoukhy et al. (2017) [33] Combined	C-ICC: 0.99 (0.99; 1.00)			A-ICC: 0.99 (0.99; 1.00)	
	Clark et al. (2013) [31]	CCC: 0.93 (0.87; 0.96)			r: 0.95 *	
	Ripic et al. (2022) [35]	C-ICC: 0.964 (0.914; 0.985)			A-ICC: 0.965 (0.918; 0.985)	
	Mentiplay et al. (2015) [36] comfortable speed		Ki: ICC 0.75 (0.53; 0.88) Vi: ICC 0.76 (0.53; 0.88)		r: 0.99	
	Mentiplay et al. (2015) [36] fast-paced		Ki: ICC 0.77 (0.54; 0.89) Vi: ICC 0.79 (0.89; 0.90)		r: 0.96	
	Dolatabadi et al. (2016) [37] usual pace		C-ICC: 0.89			
▸	Fosty et al. (2016) [38] MCBS			ICC: 0.13; 0.91		Bias: 0.013 ± 0.015 m/s
▸	Fosty et al. (2016) [38] PCBS			ICC: 0.63; 0.91		
	Otte et al. (2016) [39] Ki			ICC: 0.81 (0.67; 0.91)		
	Otte et al. (2016) [39] Vi			ICC: 0.80 (0.67; 0.91)		
▸	Kanko et al. (2021) [44]				A-ICC: 1.00 A-ICC LB: 1.00 A-ICC UB: 1.00 r: 1.00	
	Vilas-Boas et al. (2019) [46] walking towards sensor				Ki V1: r 0.87 (0.79; 0.95) Ki V2: r 0.89 (0.76; 1.00)	
	Vilas-Boas et al. (2019) [46] walking away from sensor				Ki V1: r 0.78 (0.61; 0.95) Ki V2: r 0.78 (0.55; 1.01)	
	Dubois & Bresciani (2018) [48]				ICC: 0.96	
	Muller et al. (2017) [49]				ICC (A,1): 0.999 r: 1.000 *	

▸: treadmill-based protocol, CCC: consistency correlation coefficient, r: Pearson’s correlation coefficient, ICC: intraclass correlation coefficient, A-ICC: agreement intraclass correlation coefficient, Ki: Kinect, Vi: Vicon, AP: anteroposterior, ML: mediolateral, V: vertical, FE: flexion extension, LF: lateral flexion, AR: axial rotation, LOA: limit of agreement, RMSE: root mean square error, RMSD: root mean square deviation, cm: centimeter, *: Significant (p<0.05) °: degree, LB: lower body, UB: upper body, AE: absolute error, RMS: root mean square, CMC: coefficient of multiple correlations, SD: standard deviation.

#### 2.4.2. Quantitative Analysis (Meta-Analysis) Methodology

Four articles [31,33,34,35] were retained to conduct a meta-analysis. In addition, the quantitative analysis was applied on the following gait parameters: walking speed, step length, and step time.

In order to carry out the quantitative research, articles were only selected if they used ICCs for reporting reliability (C-ICC) or concurrent validity (A-ICC). This is the only valid measure of reliability according to the COSMIN guidelines [26,27,28].

In a first step, the ICC values were transformed into a Fisher’s Z effect size and the variance of the Fisher’s Z effect size ($v_{z}$). The calculation was performed in Excel and based on the first three formulas (see Formulas (1)–(4)) below [53,54]. The Fisher transformation, also known as the Fisher Z-transformation, converts a Pearson correlation coefficient into its inverse hyperbolic tangent (arctanh) [54].

In the following Formulas (1)–(6), ICC is the intra-class correlation coefficient, r represents the Pearson correlation coefficient, and n is the number of participants included in the study.
(1)$$Fisher^{’}s\;Z_{ICC}=0.5\times \ln\frac{1+ICC}{1-ICC}$$
(2)$$Fisher^{’}s\;Z_{r}=0.5\times \ln\frac{1+r}{1-r}$$
(3)$$v_{Z_{ICC}}=\frac{1}{\begin{array}{c} n-3\\ \; \end{array}}$$
(4)$$v_{Z_{r}}=\frac{1}{\begin{array}{c} n-3\\ \; \end{array}}$$
(5)$$ICC=\frac{e^{2Z_{ICC}}-1}{e^{2Z_{ICC}}+1}=tanh\;(z)$$
(6)$$r=\frac{e^{2Z_{r}}-1}{e^{2Z_{r}}+1}=tanh\;(z)$$

The pre-calculated values (Fisher’s Z and Fisher’s Z effect sizes) were analyzed using IBM SPPS Statistics for Macintosh (version 29) to calculate an overall effect size for the gait parameters [55].

The final step in the analysis was to convert the overall effect size back to an overall ICC or Pearson value for the selected parameter. This was done using the inverse Fisher Z formula (see Formulas (5) and (6)).

The results of the quantitative research were presented in a forest plot (Figure 2) and will be discussed in the result section. The interpretation of the findings of these sections is conducted according to the same guidelines as the qualitative research [29].

### 2.5. Risk of Bias Assessment

The Cochrane guidelines [56] were used to find an appropriate tool to determine the risk of bias for the articles included. The methodological quality was assessed based on the COSMIN tool for the methodological quality of studies on measurement properties [57]. For each article, it was determined whether the following components of the COSMIN checklist were appropriate, respectively “box 6 for reliability”, “box 7 for measurement error”, or “box 9 construct validity hypothesis test”.

## 3. Results

We present the results of the systematic review and meta-analysis. After the screening according to the PRISMA guidelines, 22 studies were considered eligible.

### 3.1. Risk of Bias

For this qualitative synthesis, only the above-mentioned parts of the COSMIN tool were applicable. The risk of bias assessment was performed independently by three researchers (EA, CVB, MVDB), and consensus was reached in the case of inconsistencies. The risk of bias outcomes for each article can be found in Table 6.

### 3.2. Study Characteristics

Each study was graded on the level of evidence according to the Evidence-Based Guidelines Development (EBRO) [58]. The contribution of the articles included was established by the amount of risk of bias. Grading the level of evidence was clustered per variable (e.g., concurrent validity and accuracy in spatiotemporal parameters, concurrent validity and accuracy in kinematic variables, inter-rater reliability, inter-trial reliability, and intra-session reliability). The level of evidence was evaluated independently by three researchers (EA, CVB, MVDB), and consensus was reached in the case of inconsistencies. An overview of the certainty assessment can be found in Table 4.

### 3.3. Results Qualitative Review

#### 3.3.1. Reliability

The literature search identified nine suitable articles [31,32,33,34,35,36,37,38,39] to investigate the reliability of 3D MCBS systems compared to the gold-standard 3D MBS systems. Within these articles, 170 healthy subjects contributed to the research on the reliability of the systems, and eight patients with Parkinson’s disease were included. A total of five articles [36,37,38,39,40,41,57] examined the inter-rater reliability of both markerless and marker-based systems, while two studies [33,34] looked at the reliability between two trials. Finally, two articles [35,36] did research on intra-session reliability. All results of the nine articles [31,32,33,34,35,36,37,38,39] regarding the reliability of markerless and marker-based systems are summarized in Table 5. These results are briefly described below by reliability type and parameter.

##### Inter-Rater Reliability

Kinematic Parameters:

Overall, the ICC (95% CI) values in Table 5 indicate moderate to excellent (0.69; 0.96) reliability for the knee, good to excellent (0.85; 0.95) reliability for the hip joint, and poor (−0.39; 0.20) level of inter-rater reliability for the ankle [33,34]. These measurements are observed both in a treadmill protocol [34] and in overground walking protocols [33]. Eltoukhy et al. [33] compared kinematic parameters in both healthy and patient cohorts (PD) and found that the markerless system could consistently produce similar outcomes to the marker-based system.

Spatial and temporal parameters:

Excellent ICC values were reported for both spatial and temporal parameters for MCBS [33,34,35]. However, in the treadmill protocol, wider ranges of ICC values (0.58; 0.94) were measured for the walking speed protocols at 1.3 m/s [34]. This was also true for the study of Clark et al. [31].

While poor results were reported in kinematic ankle parameters (supra), the Arango Paredes et al. [32] study suggests excellent results for spatial and temporal parameters in the ankle joint. Walking speed was measured by Eltoukhy et al. [33], Ripic et al. [35], and Clark et al. [31]. Excellent ICC values for walking speed (>0.90) were reported in both the healthy and patient cohorts. Remarkably, the PD cohort group showed excellent values, while the healthy group showed moderate ICC values [33].

##### Inter-Trial Reliability

Kinematic Parameters:

Mentiplay et al. [36] reported moderate to good ICC values for the knee joint, moderate values for the ankle joint, and poor inter-trial reliability for the hip joint with markerless camera-based systems. Marker-based systems measured higher ICC values, respectively, for the knee (0.55; 0.91), ankle (0.68; 0.75), and hip joints (0.34; 0.55) [36].

Spatial and temporal parameters

Spatial and temporal parameters measured with a markerless camera-based gait analysis system show almost the same good to excellent values compared to a marker-based system. Only for temporal parameters has a wider range in ICC been reported in both markerless camera-based and marker-based systems [36,37].

Walking speed

For walking speed measurements using markerless camera-based gait analysis systems, ICC values ranged from 0.53 to 0.89 [36,37], while marker-based systems reported ICC values between 0.53 and 1.00 [36].

##### Intra-Session Reliability

Walking speed

The markerless camera-based systems measured ICC values between 0.63 and 0.91, while the marker-based gait analysis systems measured ICC values between 0.13 and 0.91 [38,39]. This suggests that a markerless camera-based system is comparable when measuring walking speed in both overground and treadmill protocols [38,39].

##### 3.3.2. Concurrent Validity and Accuracy

Within the included studies, nineteen [31,32,33,34,35,36,38,40,41,42,43,44,45,46,47,48,49,50,51,52] assessed the concurrent validity and accuracy of MBS and MCBS systems in gait analysis. A total of 402 healthy subjects and nineteen patients (Parkinson’s disease) were included in the studies (Table 4 and Table 5).

##### Treadmill Protocol

Kinematic

Modest to excellent Pearson correlation coefficients were reported for hip and knee joint kinematics in the sagittal plane according to Eltoukhy et al. [36] and Albert et al. [45]. In the latter study, good to excellent measurements were present in the sagittal and frontal planes. In the transverse plane, the lowest (poor to modest) Pearson correlations were measured [45]. In contrast, the studies of Pfister et al. [40] and Xu et al. [42] report different results for the hip joint in the sagittal and frontal planes. Poor correlations were found for the hip joint, but they were excellent for the knee joint, and higher errors (RMSE) were reported for different walking velocities for the knee in comparison to the hip joint in the sagittal plane [40]. The latter is in contrast with the study of Timmi et al. [41], where very low errors for knee flexion in the sagittal plane were equal for slow and fast paces, but they were slightly higher for fast walking with knee adduction in the frontal plane. RMSDs were smaller than 5.5 degrees for kinematics in the study of Kanko et al. [43], except for those that represent rotations about the long axis of the segment. The latter is consistent with what was previously reported in the transversal plane for poor results in hip and knee kinematics [45].

Poor agreement and correlations for the ankle joint were reported in the study of Eltoukhy et al. [34], as opposed to Albert et al. [45], who reported excellent Pearson correlations for the ankle joint in the sagittal plane.

Spatial and Temporal

Pearson correlations for the validity and accuracy of spatial and temporal parameters were modest to excellent in all studies [34,40,42,44]. Agreement for spatial and temporal parameters were considered modest to excellent in Eltoukhy et al. [34] and Kanko et al. [44]. However, for swing time and double support time, poor correlations (Pearson (0.20; 0.49)) were found in the study of Xu et al. [42]. Errors in temporal parameters are smaller than 0.03 s for all velocities [45].

Walking speed

Pearson correlation and agreement ICC for walking speed show excellent validity and accuracy [44]. Bias was low (0.013 ± 0.015 m/s); the higher the speed, the smaller the gaps between values [38].

##### Overground Protocol

Kinematic

The validity of the hip joint varies widely between studies and protocols, from poor to excellent in terms of correlation values [33,46,47,51,52]. Only Tanaka et al. [47] mentioned that this was significant in their study.

Moderate to excellent agreement for the knee range of motion in the sagittal plane is reported in the majority of the studies.

In all studies that measured correlations between markerless camera-based systems and marker-based systems, poor correlations for the ankle joint were found, except for Ripic et al. [52], who measured excellent validity in the sagittal plane and moderate in the frontal plane for the ankle joint.

Spatial and Temporal

Good to excellent agreement was found in both spatial and temporal values [31,33,35,48,49]. In addition, an excellent correlation between markerless camera-based and marker-based scanning techniques was eventually reported for both spatial and temporal parameters [31,35,48,49].

Walking speed

A very high level of agreement and excellent correlations were concluded in all studies that measured walking speed [33,35,48,49]. These findings were considered significant in the studies of Clark et al. [31] and Muller et al. [49].

### 3.4. Results Quantitative Analysis (Meta-Analysis)

Four articles [31,33,34,35] were retained for the quantitative synthesis. Four meta-analyses were performed on the following spatial-temporal parameters, namely walking speed (inter-rater reliability and concurrent validity), step length (reliability), and step time (reliability), with a combined total of 82 participants. Because of the small number of articles, we used the random effect model in all cases. All results can be found in Figure 2.

#### 3.4.1. Quantitative Pooling of Spatiotemporal Parameters for Inter-Rater Reliability

##### Walking Speed

The pooled data from four studies [31,33,35], as mentioned in Figure 2, suggests that there is an overall excellent inter-rater reliability with a 3D MCBS system for walking speed (n = 62, ICC = 0.97 (0.93; 0.99); heterogeneity Tau^2^ = 0.13; Chi^2^ = 7.44; I^2^ = 0.61; df = 3; *p*-value < 0.001).

##### Step Length

The inter-rater reliability of step length with a 3D MCBS system was based on two studies [34,35]. Reliability is excellent, see Figure 2 (n = 42); ICC 0.92 (0.85; 0.96); heterogeneity Tau^2^ = 0.00; Chi^2^ = 0.70; I^2^ = 0.00; df = 2; *p*-value < 0.001.

##### Step Time

The inter-rater reliability of step time was based on three studies [31,34,35] mentioned in Figure 2. The analysis with a markerless 3D motion capture system showed an excellent ICC of 0.81 (0.63; 0.91); (n = 63); heterogeneity Tau^2^ = 0.02; Chi^2^ = 26.40; I^2^ = 0.14; df = 3; *p*-value < 0.001.

#### 3.4.2. Quantitative Pooling of Spatiotemporal Parameters for Concurrent Validity

##### Walking Speed

The data from two studies [33,35] indicate that there is an excellent concurrent validity for walking speed of a 3D markerless motion capture system. (n = 41; ICC = 0.98 (0.95; 0.99); heterogeneity Tau^2^ = 0.03; Chi^2^ = 29.34; I^2^ = 0.26; df = 5; *p*-value = 0.21).

## 4. Discussion

The aim of this systematic review and meta-analysis was to compare the accuracy, concurrent validity, and reliability (inter-rater, intra-session, and inter-trial reliability) of 3D MCBS systems to the gold standard 3D MBS systems. The qualitative literature review carried out suggests that camera-based markerless systems could be considered a reliable, valid, and accurate alternative to marker-based systems in gait analysis.

Although this review indicates that, regarding the ankle and movements in the transversal plane, markerless camera-based capturing systems are not accurate and valid enough to be used in a clinical setting and need further improvement.

For the assessment of ankle kinematics, markerless camera-based systems still exhibit certain limitations in terms of reliability, concurrent validity, and accuracy [33,34,36,46,51]. Several hypotheses are described regarding the lower outcome measures reported for the ankle compared to the hip and knee in the included studies. Eltoukhy et al. [34] and Mentiplay et al. [36] suggest that the reason for these results may be due to large variations in shoe types (shoe sole, shoe height) used in the protocols, resulting in wide variability in tracking the center of the ankle [33,36]. Even with a barefoot gait analysis protocol in the Ma et al. [51] study, the authors hypothesized that the low contrast between the skin of the participants and the walking track could make it difficult to distinguish the foot contact. Vilas-Boas et al. [46] hypothesized that the lower outcomes could be the result of a greater movement of limb extremities during gait and possible interferences from infrared reflections on the floor. However, Mentiplay et al. [36] stated that they may be due to the angle computation involving three joints. Thus, less accurate joint position estimations have a larger negative effect. However, they also stated that further studies are necessary to verify if angle measurement can be improved [36].

A difference can be observed when comparing the kinematic outcome measures in the different planes of the movements. In general, poor correlations and higher errors were measured in the transverse plane for all joints, while the highest correlations can be measured in the sagittal plane [43,45,47,50,51,52]. Ripic et al. [52] hypothesized that given the choice to use unconstrained skeletal models in his study and the current availability of key point estimations in the markerless model, larger differences in the frontal and transverse planes may be expected and result in lower agreement between systems given the limited ROM in these planes [52]. In the review of McGinley et al. [17], they also stated that hip rotations clearly showed the highest error for inter-session reliability and inter-rater reliability, although they mentioned that some studies report lower error for this variable, suggesting that lower error is currently achievable.

Another interesting remark is that more variability is observed for the concurrent validity and accuracy values of the temporal parameters. Xu et al. [42] developed a hypothesis for these differences. They stated that due to the more accurate measurement of heel strike and less accurate measurement of toe-off, the temporal gait parameters that relied only on heel strike timing, such as step time and stride time, had better accuracy. The parameters that relied on both, for example, double support time and swing time, had relatively low accuracy levels and were affected by the walking speed.

Furthermore, the placement of the camera sensors varied across studies. For example, in the study by Pfister et al. [40], the camera sensor was positioned on the left side of the subject at a 45° angle to the treadmill, while Xu et al. [42] placed the sensor in front of the treadmill. Mentiplay et al. [36] also placed the sensor in the frontal plane, which implied potentially lower reliability values for the Kinect and even the MCBS kinematic results, while this placement attempted to ensure a higher accuracy result in the spatiotemporal parameters. The position of the placement of the sensor could influence the concurrent validity, accuracy, and reliability outcomes. Not only could the placement in different planes be an explanation for the differences between overground and treadmill conditions, but the distance between the sensor and subjects could be as well. This difference is approximately constant during treadmill walking but varies in overground conditions. This could be an explanation for why MCBS systems showed slightly better concurrent validity results in treadmill conditions compared to overground conditions [31,42].

Walking speed had excellent inter-rater reliability, which was consistent with the findings of the meta-analysis, good inter-trial reliability, and moderate to good intra-session reliability. Concurrent validity and accuracy values across all studies were found to be excellent for walking speed. Ripic et al. [35] stated that the results indicate that the markerless method can provide a valid measure of walking speed. Moreover, Fosty et al. [38] measured that higher walking speeds resulted in smaller gaps between values and therefore suggested better accuracy.

The meta-analysis showed excellent consistency agreement (concurrent validity) and inter-rater reliability for gait speed and step length for MCBS. Even step time could be reliably tracked with the markerless camera system compared to the marker-based systems. This finding, suggested by this quantitative synthesis, can be seen as a confirmation of what other studies have already found in their research [31,33,34,35,36].

### Limitations

Among the 22 articles included in this review, noticeably more studies examined spatial and temporal outcomes rather than kinematic outcomes of the lower limb during gait. This was also mentioned in the review of Zeng et al. [59].

The heterogeneity concerning different types of reliability (inter-rater, inter-session, and intra-session) limits the ability to make firm conclusions regarding the reliability of a markerless system compared to a marker-based system. This was also stated by the review of McGinley et al. [17]. In addition, for the concurrent validity and accuracy values, it is more difficult to draw general conclusions because of the wide range of methodological approaches that are used in the different articles.

For this systematic review, it should be acknowledged that studies with small numbers of participants were identified, which leads to inaccurate estimates. Furthermore, most studies were conducted with young, healthy individuals, which does not provide a good representation of the applications of MCBS systems in clinical settings [52]. This should be considered when interpreting data and generalizing results.

Some concerns about the risk of bias in the studies should be considered. Studies were also identified that only partially described some methodological issues, and it is important to mention that different methodological protocols were used. Therefore, it is almost impossible to generalize all these different results into conclusions. Some articles addressed the review question indirectly, raising concerns about their relevance and applicability to specific patients or settings. More research is needed to adequately compare the accuracy, concurrent validity, inter-rater, inter-session, and intra-session reliability of these markerless and marker-based gait analysis systems.

Mentiplay et al. [36] recommended that for future research, the gait analysis should be performed with standardized footwear or barefoot conditions. This should improve the ankle visualization and therefore the ankle joint kinematics. Springer and Yogev Seligmann [60] concluded in their focused review that customization and standardization of methodological procedures are necessary for future research. They also mentioned that before a markerless gait analysis system can be fully implemented in clinical use, future research involving patients with gait pathologies is required [60]. Even in this review, only two articles appeared to have looked at studying patient populations. Therefore, it is very difficult to generalize from the results of only two studies.

Another interesting consideration regarding differences in sampling rates is whether they might influence accuracy, validity, and reliability, as different protocols used different sampling rates [60]. It appears that this has not yet been investigated for accuracy, validity, and reliability between markerless and marker-based gait analysis protocols.

For the quantitative research, in terms of the articles included, only four [31,33,34,35] were suitable for statistical analysis. This was primarily because many studies employed different outcome measures that couldn’t be compared.

The repetition of measurements from the same studies in one segment of the analysis must be acknowledged as a limitation in this research. This repetition may influence the outcomes towards the findings of studies that are represented multiple times within the same analysis.

## 5. Conclusions

In conclusion, based on the included articles in this review, the results suggest that 3D MCBS can match the accuracy, concurrent validity, and inter-rater, inter-session, and intra-session reliability of spatiotemporal variables in both treadmill and overground conditions against the golden standard marker-based protocols. The outcomes of the kinematic variables of the lower limbs, more specifically the ankle joint, suggest weaker results regarding accuracy, concurrent validity, and reliability. However, it can be concluded that for both treadmill and overground conditions, the validity and accuracy of the hip and knee joints showed good to excellent results in most cases. The results of the meta-analysis confirmed these findings, although it was conducted on only three parameters (walking speed, step length, step time) for inter-rater reliability and one (walking speed) for concurrent validity. 3D MCBS are less time consuming and easier to use, and they reproduce a free natural movement of the end user without affecting her/his performances. While MBS remain the gold standard for many applications, 3D MBCS offer a promising alternative with numerous advantages. By addressing limitations associated with marker-based approaches, 3D markerless systems pave the way for more efficient, non-invasive, and ecologically valid motion capture solutions, advancing research and applications in fields ranging from biomechanics to clinical gait assessment.

## Figures and Tables

**Figure 1 sensors-24-03686-f001:**
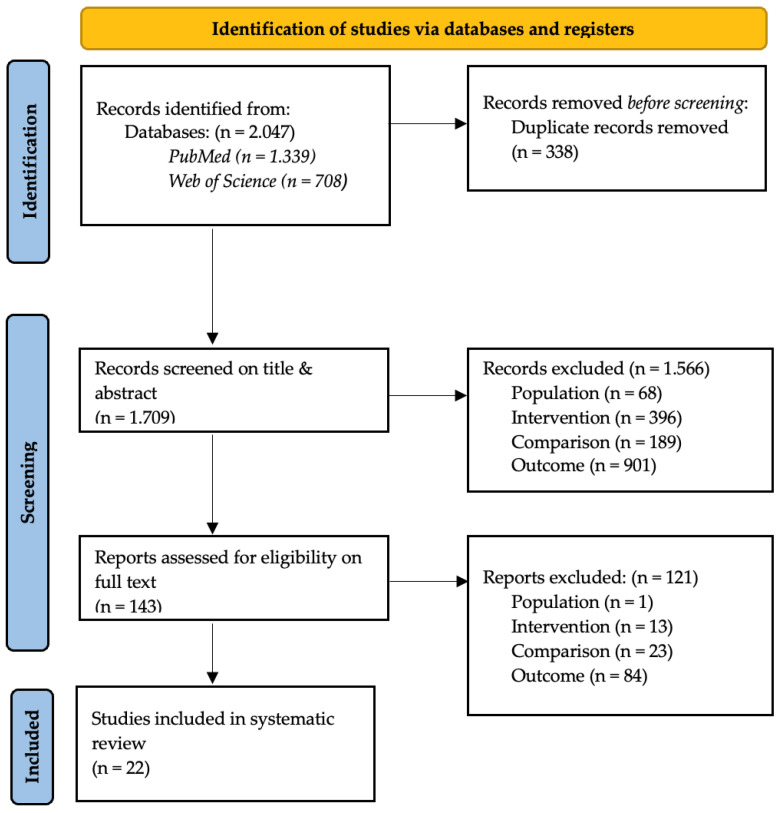
Prisma Flow Chart.

**Figure 2 sensors-24-03686-f002:**
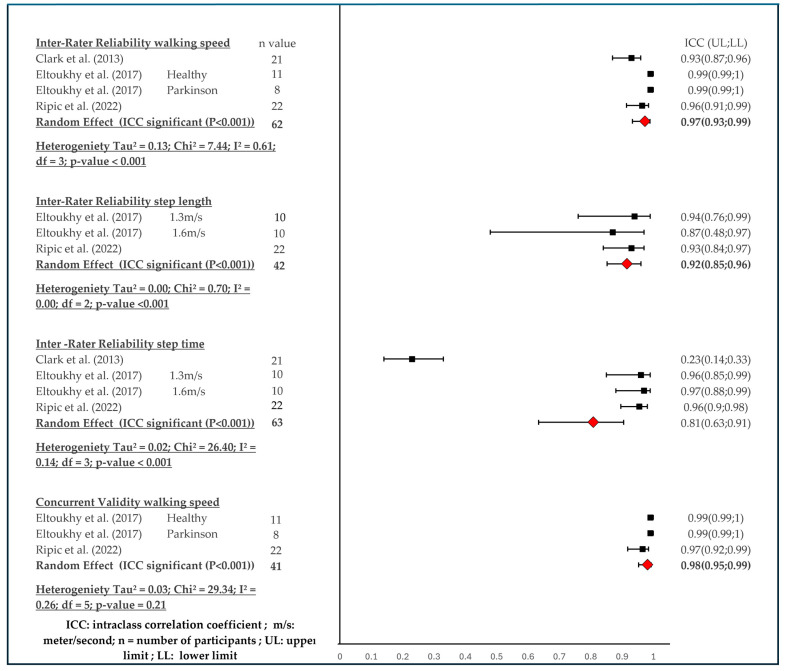
Meta-analysis data for inter-rater reliability and concurrent validity (ICC) [31,33,34,35].

**Table 1 sensors-24-03686-t001:** Database Search Strategy in PubMed.

Database: PubMed	Search Strategy
P	(Adults OR 18 years OR elderly)
	**AND**
I	(Gait OR walking)
	**AND**
C	(3D marker-based motion analysis OR 3D body scanning OR 3D capture OR three-dimensional joint angles OR 3D temporal scanner OR 3 dimensional body scanning OR three dimensional body scanning OR dynamic 3D scanning OR 4D markerless motion analysis OR 4D whole body scan OR 3D scanning technique OR 4 dimensional body shape scanning OR Kinect OR Vicon OR marker motion capture systems OR markerless motion capture systems OR dynamic movement capture OR 4D capture OR dynamic anthropometry OR motion capture OR markerless motion capture OR stereophotogrammetry OR “body shape” OR “ body scan” OR optoelectronic device OR optoelectronic system OR human motion tracking algorithm OR dynamic movement tracking)
	**AND**
O	(Concurrent Validity OR Validity OR reliability OR accuracy OR landmark position error OR correlation OR Reproducibility of results [Mesh] OR comparison OR clustering high-dimensional data))

P: Patients, I: Intervention, C: Comparison, O: Outcome.

**Table 2 sensors-24-03686-t002:** Database Search Strategy in Web of Science.

Database: Web of Science	Search Strategy
P	(ALL = (Adult OR Adults OR 18 years OR elderly)
	**AND**
I	ALL = (Gait OR walking)
	**AND**
C	ALL = (3D marker-based motion analysis OR 3D body scanning OR 3D capture OR three-dimensional joint angles OR 3D temporal scanner OR 3 dimensional body scanning OR three dimensional body scanning OR dynamic 3D scanning OR 4D markerless motion analysis OR 4D whole body scan OR 3D scanning technique OR 4 dimensional body shape scanning OR Kinect OR Vicon OR marker motion capture systems OR markerless motion capture systems OR dynamic movement capture OR 4D capture OR dynamic anthropometry OR motion capture OR markerless motion capture OR stereophotogrammetry OR “body shape” OR “body scan” OR optoelectronic device OR optoelectronic system OR human motion tracking algorithm OR dynamic movement tracking)
	**AND**
O	ALL= (Concurrent Validity OR Validity OR reliability OR accuracy OR landmark position error OR correlation OR Reproducibility of results OR comparison OR clustering high-dimensional data))

P: Patients, I: Intervention, C: Comparison, O: Outcome.

**Table 3 sensors-24-03686-t003:** Eligibility criteria following the PICO method.

Inclusion	Exclusion
Human adults (18+ years of age) Full text is written in English or Dutch RCT, clinical trial, comparative study, cohort study, cross-sectional study Gait analysis performed by a markerless camera-based 3D and 4D motion capture system of lower body (pelvis, hip, knee, and ankle), marker-based 3D motion capture systems of lower body (pelvis, hip, knee, and ankle). Gait performed overground or on a treadmill is allowed indoor, considering only walking speed <1.8 m/s.	Animals, children (<10 years of age), and adolescents (<18 years of age) Full text is written in any language other than English or Dutch Meta-analysis, (systematic) reviews Gait analysis performed by markerless camera-based 2D systems. Gait performed outdoor, stair walking, running, indoor during assessments (TUG, BBS, 10MWT, …). Measurements of gait analysis of only upper limb, trunk, or pelvis.

RCT: Randomized control trial; 3D: three-dimensional; 4D: four-dimensional; m/s: meters per second; 2D: two-dimensional; TUG: timed up and go; BBS: Berg Balance Scale; 10MWT: 10 m walking test.

**Table 6 sensors-24-03686-t006:** Summary of Risk of Bias Assessment based on COSMIN tool, (I = inadequate, D = doubtful, A = adequate, V = very good).

Study (year)	Reliability	Validity	Accuracy	Conclusion
Clark et al. [31]	X	X		D
Arango Paredes et al. [32]	X			I
Eltoukhy et al. [33]	X	X		A
Eltoukhy et al. [34]	X	X		A
Ripic et al. [35]	X	X		A
Mentiplay et al. [36]	X	X		A
Dolatabadi et al. [37]	X			A
Fosty et al. [38]	X		X	D
Otte et al. [39]	X			A
Pfister et al. [40]		X	X	A
Timmi et al. [41]			X	A
Xu et al. [42]		X	X	V
Kanko et al. [43]			X	A
Kanko et al. [44]		X		A
Albert et al. [45]		X	X	D
Vilas-Boas et al. [46]		X		A
Tanaka et al. [47]		X		A
Dubois & Bresciani [48]		X		D
Muller et al. [49]		X		A
Ruescas Nicolau et al. [50]			X	A
Ma et al. [51]			X	D
Ripic et al. [52]		X	X	A
Arango Paredes et al. [32]	X			I
Eltoukhy et al. [33]	X	X		A
Eltoukhy et al. [34]	X	X		A
Fosty et al. [38]	X		X	D
Otte et al. [39]	X			A
Mentiplay et al. [36]	X	X		A
Dolatabadi et al. [37]	X			A
Pfister et al. [40]		X	X	A
Timmi et al. [41]			X	A
Xu et al. [42]		X	X	V
Kanko et al. [43]			X	A
Kanko et al. [44]		X		A
Albert et al. [45]		X	X	D
Vilas-Boas et al. [46]		X		A
Tanaka et al. [47]		X		A
Clark et al. [31]	X	X		D
Dubois & Bresciani [48]		X		D
Muller et al. [49]		X		A
Ripic et al. [35]	X	X		A
Ripic et al. [52]		X	X	A
Ruescas Nicolau et al. [50]			X	A
Ma et al. [51]			X	D

(I = inadequate, D = doubtful, A = adequate, V = very good).

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
