# Peer review of "Accuracy, Validity, and Reliability of Markerless Camera-Based 3D Motion Capture Systems versus Marker-Based 3D Motion Capture Systems in Gait Analysis: A Systematic Review and Meta-Analysis"

_sensors, 2024, doi:10.3390/s24113686_

Round 1

Reviewer 1 Report

Comments and Suggestions for Authors

The aim of this systematic review and meta-analysis was  to compare the accuracy, validity, and reliability of Markerless Camera- 2 based 3D Motion Capture Systems versus Marker-based 3D 3 Motion Capture Systems in Gait Analysis.

Authors provide sufficient background in the introduction section.

The materials and methods section is adequately described. The systematic review methodology adheres to the PRISMA guidelines, ensuring a thorough and transparent approach to literature selection. The eligibility criteria are clearly defined.

The article effectively discusses the limitations of both markerless and marker-based systems.

There are some limitations of the study, but authors included explanation at the end of the manuscript.

Author Response

Dear reviewer,  We would like to thank you for your positive feedback and appreciation of our article.  

Reviewer 2 Report

Comments and Suggestions for Authors

- Please refrain from citing your own publications too much. 

- Tables 5 and 6 are rather inconvenient to read and they are exactly one after another, please revise the presentation of these 2 tables. A systematic review is not merely putting all of the summaries in tables. 

- What do "1 of 33", "2 of 33", etc in Tables 5 and 6 mean? If there are 33 pages of table, why only 17 pages on the manuscript?

- It is rather difficult to read the table content especially in "2 of 33", "4 of 33", "13 of 33", and "17 of 33". 

- Table 7 is of inferior quality and it is cropped out.

- A deeper analysis of the "summaries" is essential in a review paper; again, it is not just summarizing the previous studies.  

Comments on the Quality of English Language

Extensive editing is required. 

Author Response

Dear Reviewer, thanks for your comments. We provided an extensive editing of the entire manuscript. References were adapted and used were necessary. 

Regarding Table 5 and 6, the presentation was revised. However, we presented the tables according to the PRISMA guidelines (https://www.prisma-statement.org). 

Regarding the following question: “What do "1 of 33", "2 of 33", etc in Tables 5 and 6 mean? If there are 33 pages of table, why only 17 pages on the manuscript? It is rather difficult to read the table content especially in "2 of 33", "4 of 33", "13 of 33", and "17 of 33". 

The page numbers in the Sensors layout are referred to as "1 of 33" as this is not part of the table. We have modified our table to avoid any further confusion. It is a problem of layout that we corrected. 

Then regarding table 7 (see lines 447), we have modified it by improving its layout and converting it into a larger format table. 

We improved the manuscript by providing a deeper analysis according to the PRISMA guidelines. 

Reviewer 3 Report

Comments and Suggestions for Authors

The review article proposes a meta-analysis comparing marker-less and marker-based motion capture systems (camera-based in both cases) for gait analysis. The authors more specifically compared the accuracy, validity and reliability of the two types of system on kinematic and spatio-temporal gait parameters. After a two-stage screening procedure, 22 papers were included in the meta-analysis. They conclude that the accuracy, validity and reliability of marker-less systems mostly match those of marker-based systems for spatio-temporal parameters, be it over-ground or on a treadmill, whereas marker-based systems are still slightly better for kinematic variables.

The paper is clear and well-written. The authors obviously made ‘choices’ regarding inclusion criteria, but the rationale behind these choices as well as the method used are transparently reported. Overall, I think this is a very useful paper for the community (with a lot of details) and I do not have any particular reservation.

Author Response

(The authors gave the same response as above.)

Reviewer 4 Report

Comments and Suggestions for Authors

Title: Accuracy, Validity, and Reliability of Markerless Camera- 2 based 3D Motion Capture Systems versus Marker-based 3D 3 Motion Capture Systems in Gait Analysis: A Systematic Re- 4 view and Meta-Analysis

 SUMMARY

This systematic review and meta-analysis summarized the accuracy, validity, and reliability of different gait-related variables (e.g., spatiotemporal, kinematics in select planes) for 3D markerless motion capture technology compared to traditional 3D marker-based motion capture technology. 22 studies met all inclusion criteria. There was variability for all outcome variables, but in general, the psychometric properties of 3D markerless motion capture technology was acceptable for spatiotemporal data and sagittal plane hip and knee range of motion. It was not acceptable for the ankle kinematics.

SPECIFIC CONCERNS

ABSTRACT

I find the use of abbreviations excessive and hard to follow as a reader. Please reduce.

Results – the ankle angles demonstrated poor outcomes. This should be highlighted to better present an unbiased reflection of the findings.

See the comment later about providing thresholds by which the value judgment of ‘excellent’ was applied for statements, including “Walking speed was considered excellent in all cases.” For instance, Table 5 shows some ICCs for walking speed were 0.75 and 0.77 – are these excellent? Or just good? This similarly applies to the hip and knee summary. The last statement says “sagittal plane are considered valid and reliable” but earlier we acknowledged that some were “moderate.” Is “moderate” considered by most consumers as “valid and reliable”? For instance, Tanaka et al [49] reported Pearson r of [0.43,0.88] for the hip and [0.49,0.88] for the knee.  

INTRODUCTION

Define and differentiate ‘validity’ and ‘accuracy’. A reliability definition would be helpful, too.

Line 50 – define 3D or 4D markerless camera-based motion capture systems more, since 2D video was an exclusion criterion i.e., talk about the number of cameras/sensors and technology of the type of systems that were allowable.

 METHODS

It was not clear in the Introduction or Methods why 2D camera or pose estimation methods were not allowed or discussed, especially in cases where multiple cameras were used to recreate 3D estimates. That is fine if that technology is not of interest, but then at minimum, the authors should acknowledge and compare results to this other technology in the Introduction and/or Discussion (e.g., Nakano et al., 2020 PMID: 33345042).

Line 76 – Clarify if “PICO” is meant instead of PICOS.

Table 1 – justify the upper limit walking speed of 1.8 m/s.

Table 1 – some items in #4 exclusion list I believe need to be re-written because I think a double-negative is being created (i.e., first negative = exclusion, second negative = “no …”).

Table 2 – The authors reference the use of MeSH terms. For the populations of interest for this study seem to correspond to the following MeSH terms: Adult, Middle Aged, Aged. The search terms provided for “P” are not clear that all individuals 18 years or older would have been included in the search results. Please clarify.

Lines 126-147: More explanation and careful use of terms is needed. For instance, the authors should define ZICC and Zr (is one meant to represent C-ICC and A-ICC listed in lines 123? Also define r and n, as used in the equations. In Line 142, what are the differences in First Z and Fisher Z effect sizes? Also, should all “Fisher” be “Fisher’s”? I also believe equation 4 should be split into two as Summary ICC and Summary r are two separate outputs and they use different z values, as I’m understanding it. Just writing the / is confusing, as that symbol can be used to mean mathematical division.

Line 150 – Somewhere in the Methods should be the cutoff values for interpreting ICCs as poor, moderate, good, excellent, etc. as these where often mentioned in the Abstract, Results and Discussion.

RESULTS

Table 5 – Make sure angles are indicated by their plane; this was omitted for some references (e.g.,  Eltoukhy et al. [35], Eltoukhy et al. [36],…

Line 216 – Is “Overall ICC” the same as the “Summary ICC” from equation 4? If yes, does it correspond to C-ICC or A-ICC, as defined in Line 123? Please ensure very careful use of language so that the reader can follow. This applies throughout the Results and Discussion.

Line 268 & Line 300 – For these sections, make sure it is clear which plane the results apply to. Every for the reference standard of marker-based motion capture, the transverses plane is more fraught with measurement error/unreliability. I suggest even splitting this section up into results for the sagittal, coronal, and transverse planes for each interpretation. (The latter section is better, but the plane has not been specified for the hip, nor the first sentence about the ankle).

Line 281 – clarify which plane is the vertical plane.

Table 6 – I disagree that r is a measure of validity or accuracy. For instance, dataset A {1,2,3,4,5} and dataset B {10,20,30,40,50} will have a perfect r = 1.0, but there is no agreement in values between datasets.

Table 7 was cut off in the document I was able to see, so I could not fully review it.

DISCUSSION & CONCLUSION

Line 361-363: This makes too much of a sweeping statement for spatial, temporal, 3 planes of kinematics for 3 lower extremity joints. If talking about kinematics, a thorough study by discrete gait events (e.g., angle at foot contact, peak angle during swing phase) vs. overall gait (e.g., range of motion) was not investigated in this study. Delete this sentence or edit.

Line 466 – This is too strong of a statement given the variability in the different markerless systems assessed. Rephrase.

Line 471-472 – Tone down this claim too. There’s a big difference in measuring the total range of motion in the sagittal plane of gait versus accurately measuring the angle during a discrete phase of gait.

Comments on the Quality of English Language

Minor edits for English.

Author Response

Dear Reviewer, thanks for your comments. We reduced the abbreviations on the entire document (and especially in the abstract, see modification in red). While regarding the abstract we changed and clarified ankle angles (line 23). 

In addition, regarding “See the comment later about providing thresholds by which the value judgment of ‘excellent’ was applied for statements, including “Walking speed was considered excellent in all cases.” For instance, Table 5 shows some ICCs for walking speed were 0.75 and 0.77 – are these excellent? Or just good? This similarly applies to the hip and knee summary”. We specified and adapted in the article (see lines 23-29). 

INTRODUCTION 

We defined, ‘validity’ and ‘accuracy’, ‘reliability, see lines 67-80. 

We defined 3D or 4D markerless camera-based motion capture systems more (see lines 50-56, see 460-480), since 2D video was an exclusion criterion i.e., talk about the number of cameras/sensors and technology of the type of systems that were allowable. Please see lines 126-136. 

METHODS 

Please see the previous comments about 2D-3D (lines 126-136). We clarified “PICO” is meant instead of PICOS (see lines 140). Then we justify 1.8 m/s, (see lines 124-126). We modified table 1, for double negative. Regarding table 2, Table 2 – “The authors reference the use of MeSH terms. For the populations of interest for this study seem to correspond to the following MeSH terms: Adult, Middle Aged, Aged. The search terms provided for “P” are not clear that all individuals 18 years or older would have been included in the search results. Please clarify”.   

We checked these MeSH terms, but they didn't produce any additional results, so we deleted them. We took the "all fields" category because the MeSH terms are automatically included when you take a keyword, so indirectly we took the MeSH terms too. Finally, we ended up including everyone over the age of 18 in our search string. 

While “Lines 126-147: More explanation and careful use of terms is needed. For instance, the authors should define ZICC and Zr (is one meant to represent C-ICC and A-ICC listed in lines 123? Also define r and n, as used in the equations. In Line 142, what are the differences in First Z and Fisher Z effect sizes? Also, should all “Fisher” be “Fisher’s”? I also believe equation 4 should be split into two as Summary ICC and Summary r are two separate outputs and they use different z values, as I’m understanding it. Just writing the / is confusing, as that symbol can be used to mean mathematical division”. 

We changed (see lines 214-242). 

Then, “Line 150 – Somewhere in the Methods should be the cutoff values for interpreting ICCs as poor, moderate, good, excellent, etc. as these where often mentioned in the Abstract, Results and Discussion”. 

We modified it and you can find it at lines 185-189. 

RESULTS 

Table 5 – Make sure angles are indicated by their plane; this was omitted for some references (e.g.,  Eltoukhy et al. [35], Eltoukhy et al. [36]. We added the planes of the movements in table 5 and 6. 

 Line 275-277 – Is “Overall ICC” the same as the “Summary ICC” from equation 4? If yes, does it correspond to C-ICC or A-ICC, as defined in Line 123? Please ensure very careful use of language so that the reader can follow. This applies throughout the Results and Discussion. 

ine 268 & Line 300 – For these sections, make sure it is clear which plane the results apply to. Every for the reference standard of marker-based motion capture, the transverses plane is more fraught with measurement error/unreliability. I suggest even splitting this section up into results for the sagittal, coronal, and transverse planes for each interpretation. (The latter section is better, but the plane has not been specified for the hip, nor the first sentence about the ankle). Line 281 – clarify which plane is the vertical plane.  

We changed and adapted the entire document for the planes. Please also see lines 348-366. Table 6 and 7 were modified. 

DISCUSSION & CONCLUSION 

Reagrding “ Line 361-363: This makes too much of a sweeping statement for spatial, temporal, 3 planes of kinematics for 3 lower extremity joints. If talking about kinematics, a thorough study by discrete gait events (e.g., angle at foot contact, peak angle during swing phase) vs. overall gait (e.g., range of motion) was not investigated in this study. Delete this sentence or edit. 

Line 466 – This is too strong of a statement given the variability in the different markerless systems assessed. Rephrase. 

Line 471-472 – Tone down this claim too. There’s a big difference in measuring the total range of motion in the sagittal plane of gait versus accurately measuring the angle during a discrete phase of gait”. 

We extensively modified the discussion and the conclusion. 

Round 2

Reviewer 4 Report

Comments and Suggestions for Authors

SUMMARY

This systematic review and meta-analysis summarized the accuracy, validity, and reliability of different gait-related variables (e.g., spatiotemporal, kinematics in select planes) for 3D markerless motion capture technology compared to traditional 3D marker-based motion capture technology. 22 studies met all inclusion criteria. There was variability for all outcome variables, but in general, the psychometric properties of 3D markerless motion capture technology was acceptable for spatiotemporal data and sagittal plane hip and knee range of motion. It was not acceptable for the ankle kinematics.

GENERAL CONCERNS

Be very clear with words/phrases and their meaning. This is crucial since the authors are reporting so many types of psychometric properties for so many outcome variables and often more than one independent variable. This will be helpful to the reader, because it’s challenging to verify claims and values stated in text because I don’t know where specifically they came from in the Tables.

Relatedly, make sure the tone/confidence level of a claim is proportional to the level of evidence available. It is perfectly acceptable and is the responsibility of the authors to objectively report study findings and conclusions, which may mean saying there is insufficient evidence to draw a conclusion.

Except in the title, I suggest changing all “validity” everywhere else to “concurrent validity” since that is the type of validity assessed in this study.

Similarly, there are several spots where “markerless” was omitted before “camera-based systems.” Please add markerless back in everywhere so it’s clear to the reader (since traditional marker-based motion capture systems also use cameras).

Remove the word “invasive” as this is reserved for medical procedures “that invades (enters) the body, usually by cutting or puncturing the skin or by inserting instruments into the body.” That is not an accurate adjective for 3D marker-based methods.

SPECIFIC CONCERNS

ABSTRACT

ICC needs to be spelled out the first time it’s mentioned.

This is an example of mixing so many outcome variables in a sentence and providing ICCs values that I can’t be sure where they came from “The meta-Analysis of the reliability and validity of walking speed, step time, and step length resulted in an excellent ICC (0.80;0.92).”

INTRODUCTION

Line 52 – a better description is required in the Introduction besides just saying “3D and 4D scanners,” since ‘scanner’ is too generic. Describe the number of cameras/sensors and technology of these systems. Some from lines 126+ may be moved/repeated here.

Line 69-80 – Thank you for providing definitions of the psychometric properties, since these can be defined differently depending on context (e.g., continuous vs. count data) and/or be used as synonyms or special instances for each other (particularly accuracy and validity). I do not have access to reference 23, but from my understanding, you are using ‘accuracy’ as the exact/absolute agreement of values from two systems, whereas with concurrent validity you’re looking at the relative agreement of values from two systems. Therefore, the statistics described in section 2.4.1 should reflect these differences.

Line 74 – Only these 3 types of reliability are listed here but more are presented & discussed; update this list in Intro. Clarify it/how intra-session and inter-trial reliability differ.

Line 76 – spell out ICC the first time used in the main text.

METHODS

Lines 181-183 and 192-196 – Split up accuracy and validity in the presentation of results (i.e., have 3 tables, 1 for each psychometric property OR have 2 column in Table 6 with 1 column for accuracy and 1 column for validity results) and in describing which metrics are used to represent the 3 psychometric properties (accuracy, validity, reliability [with 3 types of reliability]). They are distinct. Doing so will help guide your reader to understand if the interpretation of whether the markerless systems are adequate or inadequate for the 3 psychometric properties.

Line 186 – I believe ‘Consistency Intra Class Correlations (C-ICC)’ should be Consistency Intra-class Correlation Coefficients (C-ICC).

Line 196 – Thank you for providing thresholds used for interpreting the reliability metrics. Please do the same for the validity and accuracy metrics, since the Results and Conclusions of this paper include phrases like “The accuracy and validity of [variable N] was considered excellent in all cases, with only a small bias.”

Line 197-198 – Fleiss, 1986 should be cited instead of reference 30 for Pearson r. Please verify Fleiss is appropriate for CCC since it’s different than Pearson r since it also measures agreement. For instance, one reference I found said that CCC <0.9 is poor while another said >0.8 is excellent (PMID: 30191186).

Lines 214-220: If I’m understanding correctly, explicitly state Fisher’s Z effect size (Fisher’s ZICC) and Fisher’s z-transformation (Fisher’s Zr).

Line 219 – I believe it should be ‘intra-class correlation coefficient’ instead.

Line 226-231 - Does the r in equation 2 mean the same thing as the r in equation 4? If not, use different letters and make sure to describe. Based on the definition in lines 238-240, I think the r on the left side of the equation is supposed to be some type of ICC, but I can’t be sure.

Line 231 – Z is not defined as used in equation 4. Is it ZICC? Zr? Or a whole new Z?

Line 235 – “Fisher’s Z” is now introduced. Is this a new variable not defined earlier or listed in equations 1-4? Or Should it be Fisher’s Zr?

RESULTS

Throughout: Because there were 4+ types of reliability measured, specify which type(s) each time a result is given. In general, inter-rater reliability or test-retest reliability are larger than inter-trial reliability, so we cannot pool all results together.

Tables – it’s unclear to the reader when there’s multiple Gait Parameters for a given domain (e.g., Temporal, Spatial) what the 1 statistical Result means. For example, ref [40] Table 6 has 5 temporal gait parameters, but only A-ICC is presented as 0.834;0.951. Are they the two values the 95% CI LL and UL? If yes, of which gait parameter?  

Table 5 – spell out 3DMA the first time used. Or, if using it as a synonym with MCBS which is an abbreviation already used, change 3DMA throughout (3 uses).

Table 6 – Please indicate somewhere in the Table whether the walking trials for the Measuring method and Reference method were collected concurrently. This would be the strongest study design method versus data from the two systems being recorded asynchronously. Comment on this in the Discussion.

Table 7 - Provide a description of the type of reliability (inter- or intra-what). “Comfortable” is spelled incorrectly. Also consider adding an indicator whether the studies were overground or treadmill walking, since that factor seems to affect outcomes.

As I’m reconsidering all the data and try to make it easier for the reader to take the info in Tables 5 & 6 and be able to draw their own conclusions and/or see if they agree with the conclusions drawn by the authors, you may consider shuffling tables around. Maybe have 1 generic table with the first 6 columns of data from Tables 5 & 6. Then create a new table where the 1st column is “Study, year” and have rows as each outcome variable (e.g., step length, step time, walking speed, sagittal ROM of the hip, frontal ROM of the hip) and have columns for the general psychometric properties (e.g., Accuracy, Validity, Inter-rater reliability, Inter-trial reliability, Intra-session reliability). Then within each cell you provide the name of the specific statistic along with its value. For studies where there’s 2 conditions (e.g,. 2 walking speeds, 2 systems, 2 populations), you could make & label 2 rows for each outcome variable.

DISCUSSION

Line 460 – modify this sentence to more accurately reflect the results: “…could be considered a reliable, valid, and accurate alternative to marker-based systems for some gait analysis variables.”

The new 2nd through 5th paragraphs of the Discussion do not discuss any of the authors’ findings, so it seems more appropriate for an Introduction (which some is repeated from the Intro, but in more elaborate form). This information can be kept, in brief, in the Discussion, but it should come after/integrated with the discussion of the study findings.

Line 503-520 – These paragraphs needs to be clear about 1) which psychometric variable(s) the authors are talking about and 2) which measurement system they’re talking about (namely for McGinley ref 17, Xu ref 47).

There is nothing in the Discussion or Limitation paragraph that acknowledges kinematic range of motion were the type of kinematic variables, which would be grossly inadequate for clinical use. They are mostly interested in angles during specific phases of gait. This needs to be addressed in the Discussion and/or Limitation.  

In the Discussion, there should be more interpretation of the different types of reliability. Was sufficient evidence available in the studies to draw conclusions for each type? Are all types of reliability the same?

CONCLUSIONS

Having 2 paragraphs each starting with “In conclusion” should be reworked to be more effective. Similarly, the summaries are too sweeping & do not acknowledge the limitations or caveats of the findings discussion just previously in the Limitations section. Namely, that there seems to be differences for treadmill vs overground, these are mostly healthy populations, kinematics include only total range of motion, and several studies showed risk of bias. Your results fully support this statement in the Limitations, “More research is needed to adequately compare the accuracy, validity, and reliability of these markerless and marker-based gait analysis systems.” Yet this statement is not reflected in 1) the Abstract, or 2) the Conclusions. Please ensure all statements and conclusions are supported by empirical evidence, with the appropriate caveats disclosed for the reader. Most statements are overly confident given the level of evidence.

Author Response

Please find the response in attachment.
